

# Transport of Pollution to a Remote Coastal Site during Gap Flow from California's Interior: Impacts on Aerosol Composition, Clouds and Radiative Balance

A. C. Martin[1*] and G. C. Cornwell[2*], S. A. Atwood[3], K. A. Moore[2**], N. Rothfuss[4], H. Taylor[4],

P. J. DeMott[3], S. M. Kreidenweis[3], M. D. Petters[4], and K. A. Prather[1,2]

[1]Climate Atmospheric Science and Physical Oceanography, Scripps Institution of Oceanography, La Jolla, CA, USA

[2]Department of Chemistry and Biochemistry, University of California San Diego, La Jolla, CA, USA

[3]Department of Atmospheric Science, Colorado State University, Fort Collins, CO, USA

[4]Department of Marine Earth and Atmospheric Sciences, North Carolina State University, Raleigh, NC, USA

*Correspondence to:* K. A. Prather (kprather@ucsd.edu)

**Abstract**

During the CalWater 2015 field campaign, ground-level observations of aerosol size, concentration, chemical
composition, and cloud activity were made at Bodega Bay, CA on the remote California coast.  A strong anthropogenic influence on air quality, aerosol physicochemical properties, and cloud activity was observed at Bodega Bay during periods with special weather conditions, known as Petaluma Gap flow, in which air from California's interior is transported to the coast. This study applies a diverse set of chemical, cloud microphysical and meteorological measurements to the Petaluma Gap flow phenomenon for the first time. It is demonstrated that
the sudden and often dramatic change in aerosol properties are strongly related to regional meteorology and anthropogenically-influenced chemical processes in California's Central Valley. In addition, it is demonstrated that the change in airmass properties from those typical of a remote marine environment to properties of a continental regime has the potential to impact atmospheric radiative balance and cloud formation in ways that must be accounted for in regional climate simulations.

## 1   Introduction

The remote northern California coast experiences a Mediterranean climate (Aschmann, 1973, Lentz and Chapman, 1989) and warm, dry summers. The vast majority of yearly precipitation occurs during winter (Regonda et al., 2005), when the north Pacific extratropical storm track extends southward and brings periodic pressure falls and
rain to the region. (Gyakum et al., 1998). Also during the winter months, conditions known as channeled gap flow can transport airmasses from much further inland to the remote coast. These episodic periods result when very low near-surface buoyancy and an onshore-directed gap-parallel pressure gradient co-occur in one of several prominent

*These authors contributed equally to this work.

**Now at School of Chemistry, University of St. Andrews, St. Andrews, UK.




gaps in the coastal mountain ranges (Overland and Walter, 1981; Neiman et al., 2006; Loescher et al., 2006; Colle et al., 2006). One such prominent gap is located near the town of Petaluma in Sonoma County, CA and can act to channel air from the North San Francisco Bay Area (SFBA), the Sacramento River Delta and California's Central Valley (CV) to coastal Northern California (see schematic in Figure 1, Neiman et al., 2006 – hereafter N06). N06

described the regional weather patterns and lower tropospheric dynamic meteorology associated with Petaluma Gap Flow (PGF) using 62 cases observed during the multi-winter deployment of a 915 MHz wind profiling radar to Bodega Bay, CA. N06 describe PGF as a near-surface shallow (~500m) stably-stratified quasi-Bernoulli flow which can lead to increased static stability, increased density, lower relative humidity and increased anthropogenic pollutants near Bodega Bay and offshore.

Evidence presented in N06 for the proliferation of anthropogenic pollutants at the coast during PGF included horizontal coast-normal transects and low-troposphere vertical profiles of Carbon Monoxide (CO) mixing ratio from a trace gas analyzer on-board just one research flight of the National Oceanographic and Atmospheric Administration (NOAA) P-3 aircraft. During the coast-normal transect CO mixing ratio doubled from 120 ppbv to 240 ppbv across a 20 km wide gradient that was located approximately 75 km offshore of Bodega Bay and Point

Reyes, CA. It was inferred that when the near-surface airmass during PGF episodes travelled from the polluted Central Valley region before arriving at the coast, the airmass would acquire properties commensurate with combustion, transportation, agriculture and manufacturing (e.g. the observation of elevated CO concentration).

When transport occurs, PGF should cause large measurable impacts on the coastal environment via an abrupt but significant change in trace gas and aerosol chemistry. Expected impacts include:

• An increase in absorption of solar shortwave radiation by black carbon aerosol, which have much greater emission sources on the continental side of the Petaluma Gap. An increase in black carbon mass may also be associated with more freshly emitted soot. Together, these factors may lead to a relative decrease in both externally mixed and internally mixed organic to elemental carbon ratios. (e.g. Chung et al., 2012; Cahill et al., 2012).

• A brightening in near-shore marine stratocumulus clouds through the cloud albedo indirect effect (Twomey, 1977; Solomon, 2007), since the inferred PGF airmass contains more numerous pollution aerosol particles, a portion of which will serve as cloud condensation nuclei.

           • Increased deposition of nitrogen containing particulate matter to the local ecosystem which may lead to increased eutrophication along the coastal shelf (Paerl, 1995; Paerl, 1997), because

particles transported during PGF may have formed or have been aged in a nitrous-oxide and ammonia enriched environment (Seinfeld and Pandis, 2012).

As part of the CalWater-2 experiment (Lueng et al., 2014; Ralph et al., 2015), measurements of trace gas concentrations, aerosol physicochemical properties and lower tropospheric meteorology were taken at the University of California, Davis Bodega Marine Laboratory during January, February and March, 2015. Using this

dataset, described in section 2, we report the abrupt changes in trace gases and particulate matter observed during five PGF events and establish composite aerosol size distributions, aerosol chemical sources, trace gas concentrations, and cloud condensation nuclei activation curves. We also identify particle aging through the





accumulation of ammonium and nitrate during PGF using detailed single particle mass spectrometry measurements. The analysis methods presented in section 3, and their results, presented in section 4, verify the above hypotheses and present a nuanced picture of the secondary heterogeneous chemistry active in aerosol particles that travel to the coast in the PGF airmass. Fine details of particle aging are location specific, but

conclusions drawn from the increase in aerosol number, changes in aerosol source, brightening of marine clouds and impact on aerosol absorption are generally applicable to many other coastal regions that periodically experience channeled offshore flow through a mountain gap.

## 2    Data Sources

### 2.1 Bodega Marine Laboratory

Measurements and samples were collected January 14, 2015 to March 9, 2015 at Bodega Marine Laboratory (BML, 38° 19'N, 123° 4') in Bodega Bay, California. BML is located south-southwest of the northern California coastal mountain ranges and north of Point Reyes National Seashore.

The sampling site at BML included two instrumented trailers located ~100 m ENE of the seashore, and ~30 m north of the northernmost BML permanent building. The Beta Attenuation Monitor (BAM), for measuring

particulate matter mass for particles smaller than 2.5 µm ($PM_{2.5}$), and IMPROVE (Eldred et al., 1997) filters for collecting $PM_{2.5}$ and particles smaller than 10 m ($PM_{10}$) were placed at an ancillary site ~15 m Southwest of the trailers. The IMPROVE samples are not used in the analysis that follows. Aerosol composition and ice nucleating particle concentrations were measured in the trailer owned by the California Air Resources Board (CARB) and operated by the University of California, San Diego (UCSD). Aerosol sizing, gas phase tracer concentrations, black

carbon mass, cloud condensation nuclei concentrations were measured in the trailer owned by the National Park Service (NPS) and operated by Colorado State University (CSU). Ambient aerosol particles were collected nearby using filters for subsequent laboratory measurement of drop freezing temperature. A more extensive description of these instruments and the data processing, quality control and archiving is given by Petters et al.[1] A meteorology station operated by the Earth System Research Laboratory, National Oceanic and Atmospheric Administration

(NOAA/ESRL) was located ~100 m North of the two trailers. The NOAA/ESRL 449 MHz wind profiling radar, radio acoustic sounding system (RASS), and 10m surface meteorology tower were operated during the two study periods and are used for the analysis presented in later sections. Descriptions of these instruments and the NOAA/ESRL Bodega Bay meteorology station (BBY) can be found in White et al. (2013).

### 2.2 Aerosol Composition

Size-resolved aerosol composition at Bodega Bay was measured with an aerosol-time-of-flight mass spectrometer (ATOFMS) and an ultrafine aerosol-time-of-flight mass spectrometer (UF-ATOFMS). The UF-ATOFMS used a diluting stage with an approximate dilution of 5x to increase aerosol ionization efficiency during periodic high particle concentrations. The use of these instruments in tandem allows the direct measurement of single-particle

---

[1] Petters, M. D., Taylor, H. P., Attwood, S., Kreidenweis, S. M., DeMott, P. J., Rothfuss, N. E., Prather, K. A. Martin, A.: Aerosol characteristics in and around landfalling Atmospheric Rivers and marine air intrusions observed during the Calwater 2015 campaign.



composition for particles in the aerodynamic diameter range 0.15-3.0 µm. The design and operating principles of these instruments have been described elsewhere (Gard et al., 1997; Su et al., 2004), thus we provide only a short overview here. Particles were dried prior to introduction to the instrument with silica-diffusion dryers in order to improve the ionization efficiency and thus improve the acquired spectra quality. Particles enter the vacuum system

through either a converging nozzle or an aerodynamic lens, wherein they are accelerated to their terminal velocities. After reaching this terminal velocity, they enter the sizing region where travel through two continuous wave laser beams (diode-pumped, Nd:YAG at 532 nm) separated by a vertical distance of 6 cm, and oriented orthogonal to each other. Because the distance is known, particle velocity can be determined by measuring the time difference between the two scattering signals. These velocities can be converted to vacuum aerodynamic

diameter ($D_{va}$) through a calibration curve generated with polystyrene latex spheres (PSLs). The velocity is also used to calculate the time when a particle will arrive in the ion source region. Upon arrival in the source region, a 266-nm Nd:YAG laser is triggered to fire upon the particle, desorb it, and generate positive and negative ions whose mass spectrum are measured with dual-polarity time-of-flight mass spectrometer.

**2.3 Size Distributions**

Aerosol size distributions at BML were measured using a scanning mobility particle sizer (SMPS, TSI Inc. Model 3936) and an aerodynamic particle sizer (APS, TSI Inc. Model 3321). The SMPS was operated with a pump flow of 0.3 lpm and a sheath flow of 3.0 lpm so that the dynamic size range varied from 13.6 to 736.5 nm. The APS operated with a sample flow rate of 1.0 lpm and a sheath flow of 4.0 lpm. The APS was externally calibrated using Polystyrene Latex Spheres (PSLS). Particles were not dried prior to sizing, but the relative humidity (RH) of the

sample line was monitored with a RH sensor (Vaisala, HMP110). Assuming spherical particles, the measured modibility diameter ($D_m$) is equivalent to physical diameter ($D_p$). APS measurements were adjusted from aerodynamic diameter ($D_a$) to $D_p$ assuming spherical particles and an effective density of 1.8 g cm$^{-3}$. Both APS and SMPS size distributions were combined to ten-minute mean distributions from their operational 1 and 5 minute scan frequency.

**2.4 Aerosol Mass**

PM$_{2.5}$ mass was determined using a beta attenuation monitor (BAM, Met One Instruments Inc., Model BAM 1020). The mass was recorded hourly. Black carbon (BC) mass concentration and attenuation were determined with a 7-channel aethalometer (Magee Scientific Corp., Model AE16-ER-P3-F0), operating in AE-30 mode.

**2.5 Cloud Activation Properties**

Size-resolved cloud condensation nuclei concentrations were measured using a streamwise cloud condensation nuclei counter (Droplet Measurement Technologies Inc., CCN-100) coupled with an SMPS. The SMPS (TSI 3080 long column) was operated at a sheath-to-sample flow rate of 5 (L min-1)-to (1.3 L min-1). Raw counts were recorded in Labview and inverted as described previously (Nguyen et al. 2014; Petters et al. 2009). The inversions account for temperature and pressure changes inside the DMA, contribution of multiply charged particles and

losses through the inlet system. The CCN was operated at a sample flow rate of 0.3 L min-1 and sheath-to-sample flow ratio of 10:1. Supersaturation is calibrated using atomized, dried ammonium sulfate aerosol (Christensen and Petters 2012). The control software cycles through an automated 12 point sequence varying supersaturation



between 0.06% and 0.67%. Activation diameters are obtained for normalized activation curves (Petters et al. 2009) and the apparent hygroscopicity parameter at standard state, ⬚ is calculated from the supersaturation and activation diameter (Christensen and Petters 2012; Petters and Kreidenweis 2007).

**2.6 Gas-Phase Measurements**

Concentrations of gas-phase pollutants were determined using a suite of gas-phase instruments, collocated with the aethalometer, CCN counter, and the sizing instruments. A $NO$-$NO_2$-$NO_x$ analyzer (TEI Inc., Model 42C), ozone analyzer (TEI Inc., Model 49C), sulfur dioxide analyzer (TEI Inc. Model 43C), and carbon monoxide monitor (Horiba Inc., APMA-370) were all utilized in this study. Gas phase measurements were recorded every second and converted to 10-minute mean time resolution.

**2.7 Remotely Sensed Cloud Properties**

Level 2 MODIS cloud products (Platnick et al., 2003) are used to estimate the range of marine stratocumulus cloud optical depth offshore from BML during PGF episodes with clear sky above low clouds, and to verify that the clouds in near-shore MODIS scenes are low-level cumulus or stratocumulus clouds.

**3    Methods**

**3.1 Method of Compositing by Measurement Period**

Composite aerosol size distributions, trace gas and aerosol type concentrations, indicators of secondary chemical aging, and CCN activation spectra corresponding to PGF periods and control periods are derived as a primary tool for addressing the hypotheses posed in this study. Herein, we define a control period to be any hourly period which does not fit the N06 definition of PGF and does not occur during short-lived episodes of concentrated local

anthropogenic pollution. Observed causes of local anthropogenic pollution included nearby brush fires, vehicle activity at BML, and "seabreeze resampling". During the latter, high concentrations of anthropogenic pollution from either a local source or from further inland that was previousy transported offshore returned to the measurement site via the afternoon sea-breeze. Since the polluted airmass may have up to 18 hours of modification by the near BML marine environment just before seabreeze resampling episodes, these were classified as local

anthropogenic pollution and were removed from the PGF and control period composites.

We followed the methodology of N06 in identifying Petaluma Gap flow using the BBY 449 MHz vertically profiling radar and 10 m anemometer (see section 4a in N06). If the 449 MHz radar and 10 m wind observations met all necessary conditions from N06, we declared the period meteorological PGF (mPGF). It is important to note that while 449 MHz wind profiles are collected hourly, all other data from the study are collected

more frequently, therefore we classified local conditions in hourly intervals.

To choose local conditions based on an indicator of anthropogenic pollution, we examined CalWater 2015 observations of CO, $NO_x$, SMPS number concentration integrated from 13.6 nm to 736.5 nm (CN), and $PM_{2.5}$ (collectively, peripherals). During mPGF, the interquartile range of CO, and $NO_x$ was higher than the interquartile range of the same measurements during periods that did not fit mPGF (not shown). In addition, for indicators of

fine particulate concentration (CN, $PM_{2.5}$), the median value during PGF is higher than the upper quartile value during all periods. For all anthropogenic pollutant indicators, the maximum observation and much of the upper



quartile range is higher than any observation taken during mPGF, and occurred during local anthropogenic pollution periods. An example is shown in Figure 1, which contrasts lower tropospheric horizontal wind and virtual potential temperature ($\theta_v$), selected peripheral measurements and EC and SSA particle subtypes (see methods for particle typing, later this section) during a period when neither mPGF was observed nor anthropogenic pollutants were high (Fig. 1a), a period when it is suspected local anthropogenic pollutants were sampled (Fig. 1b) and a mPGF period (Fig. 1c). Note that during the local pollutant episode, CN and CO were elevated to the same levels as during the mPGF period for a few hours. The onset of the elevated pollutant period occurs near the maximum in onshore sea-breeze (NNW 200 m wind near 03 UTC on February 12[th]). Pollutant concentrations decrease again a few hours after local sunset, when the offshore land-breeze become established. The period February 12, 2015 @ 0300 UTC through February 12, 2015 @ 11 UTC is an example of a seabreeze resampling period.

In order to exclude local or sea-breeze resampled anthropogenic pollutants from control periods (hereafter CTL), we imposed an additional requirement based upon CO concentration - hourly mean CO concentration must be above the CalWater 2015 mean plus two standard deviations (138.1 ppbv). Along with mPGF, this requirement forms the basis of a decision table (Table 1) that allows the separation of CalWater 2015 measurements into 4 composites. We choose CO concentration as our additional discriminator because its interquartile range during mPGF is entirely above the interquartile range from all other periods, because it's overall variability is the lowest among peripheral measurements and because elevated near-surface CO concentration was observed by aircraft during a PGF event reported in N06. Table 1 allows the compositing of observational period by PGF (mPGF and elevated CO conditions met), CTL (neither mPGF nor elevated CO conditions met), LOCAL (mPGF not met, elevated CO met), and DIFFUSE (mPGF met, elevated CO not met). In this light, Figs. 1a, b and c can be seen as examples of CTL, LOCAL and PGF periods, respectively.

### 3.2 Derivation of Angstrom Absorption Exponent from Aethalometer Observations

The attenuation recorded by the aethalometer was used to derive the aerosol absorption coefficient ($\sigma_{ATN}$) at the instrument's native 5 minute resolution following the method described in Collaud Cohen et al. (2010) (see their equation 2). This value was not corrected, and thus not reported directly, as techniques for correcting aethalometer require a coincident multi-channel measurement of aerosol scattering, and this additional measurement was not taken during CalWater 2015. It is noted that values of $\sigma_{ATN}$ generally fall in the range reported by previous studies in continental regions (e.g. Table 1 in Chung et al., 2012). The hourly mean aerosol absorption coefficient in the channels 470 nm, 520 nm, 590 nm, and 660 nm were used to derive the Angstrom absorption exponent (AAE), using the relation $\sigma_{ATN}(\lambda) = C_0 \lambda^{-AAE}$. AAE is calculated by regression to the uncorrected $\sigma_{ATN}(\lambda)$. Note that Weingartner et al. (2003) estimated that errors in $\sigma_{ATN}$ are only a very weak function of wavelength in the channels used, thus it is expected that instrument errors do not contribute significantly to the estimate of AAE.

### 3.3 Assigning Particle Type to ATOFMS Spectra

ATOFMS and UF-ATOFMS can provide information on size and chemical composition (via mass spectra) for an individual particle. Generally, positive spectra reveal particle source while negative spectra provides information on the atmospheric processing a particle has undergone (Guazzotti et al., 2001; Sullivan et al 2007; Prather et al., 2008). ATOFMS, but not UF-ATOFMS spectra were filtered for periodic radio frequency interference caused by





a sub-optimally operating instrument component. A total of 115,416 particles were scattered and hit during PGF events, and 1,835,387 during the control time periods (see section 3 for definition of PGF and control periods).

Single particle spectra and size data were loaded into Matlab (The MathWorks, Inc.) and analyzed via the software toolkit YAADA (http://www.yaada.org/). Particles were divided into clusters based on their mass spectral features

via an adaptive neural network (ART-2a, vigilance factor 0.8, learning rate 0.05 and 20 iterations regroup vigilance factor of 0.85) (Rebotier and Prather, 2007; Song et al., 1999). Greater than 95% of art2-a analyzed particles were grouped were recombined into 11 types based upon their characteristic mass spectra and size distributions. Similar to previous field studies using ATOFMS (Sullivan et al., 2007; Pratt et al., 2009; Cahill et al., 2012, Qin et al., 2012), particle types were assigned by a human operator based upon similarity to known types from previous field

and laboratory studies. Calculated standard error in particle fraction were less than 1% for all particle types and thus were not included in the discussion. These results are summarized in Table 2.

**3.4 Determining Aging Mechanisms Using ATOFMS**

It is important to describe not only particle sources, but also secondary aging impacts as the aging mechanism will change the light absorption cross-section of carbonaceous aerosols. For instance, a sulfate coating can increase the

light-absorbing properties of soot by a factor of 1.6 (Moffett and Prather, 2009). Internally-mixed EC and OC have greater absorption profiles compared to homogeneously mixed particles of either species. (Schnaiter et al., 2005). Additionally, aging can increase particle hygroscopicity through condensation and reaction of gases like $NO_x$ or $SO_2$ (Wang et al., 2010; Mochida et al., 2006; Zuberi et al., 2005; Zhang et al., 2008) or oxidation of carbonaceous species (Zuberi et al., 2005; Kotzick et al., 1997). Increased particle hygrospocity can increase the CCN activity

of particles and their growth factor, profoundly impacting radiative effects   Finally, accumulation of nitrogen species on particles can lead to increased deposition of nitrate and ammonium and impact oceanic biology (Paerl, 1995; Paerl et al., 1997).

The ATOFMS is a powerful tool to measure particle aging because of its ability to measure single-particle composition and directly determine the type and extent of particle aging.. For similar particles of the same type,

relative peak areas (RPA) qualitatively reflect the amount of a species on a particle in relation to other species (Bhave et al., 2002; Gross et al., 2000; Prather et al., 2008) and thus can be used to investigate the mechanism of aging (Cahill et al., 2012). During this study, the mixing state of single particles with secondary markers was investigated by identifying and comparing peak areas for ammonium ($^{18}NH_4^+$), amines ($^{58}C_2H_5NHCH_2^+$, $^{59}NC_3H_9^+$, $^{86}(C_2H_5)_2NCH_2^+$), sulfate ($^{97}HSO_4^-$, $^{195}H_2SO_4HSO_4^-$), nitrate ($^{46}NO_2^-$, $^{62}NO_3^-$, $^{125}H(NO_3)_2^-$), elemental carbon ($^{12}C^+$,

$^{36}C^+$, $^{60}C^+$), and organic carbon ($^{27}CHN/C_2H_3^+$, $^{29}C_2H_5^+$, $^{37}C_3H^+$, $^{43}C_2H_3O^+/CHNO^+$). Other markers of heterogeneous processing were investigated, but no notable patterns emerged. For this anlaysis, a particle was considered to be an internal mixed with a species if it had an RPA greater than 0.5% for the characteristic ion markers, similar to the methodology employed by Cahill et al. (2012).

**3.5 Estimates of Cloud Droplet Number Concentration and Marine Stratocumulus Albedo Change**

Size distribution, hygroscopicity, and CCN concentration measurements were collated from periods classified as PGF and CTL. Cumulative CCN supersaturation spectra, defined as median CCN concentration as a function of supersaturation were constructed from the integrated CCN and size distribution data. The spectra were fit to a two





mode hypergeometric model (Cohard et al., 1998). to estimate cloud droplet number concentration for a range of updraft velocity.

The albedo change ($\Delta A_c$) of near-shore marine stratocumulus during PGF episodes was determined using equation 7 in Platnick and Twomey (1994):

$$5 \quad \Delta A_c = \left[ A_C (1 - A_C) \left( \chi^{\frac{1}{3}} - 1 \right) \right] \left[ A_C \left( \chi^{\frac{1}{3}} - 1 \right) + 1 \right]^{-1},$$

where $\chi = N_{PGF}/N_{CTL}$ is the ratio of CDNC in PGF conditions to CDNC in CTL periods. This analytical formulation relies upon the assumptions of conservative scattering, nearly invariant asymmetry parameter, and constant liquid water path. Furthermore, the estimate we present herein of albedo change during PGF episodes assumes that marine stratocumulus clouds are present near Bodega Bay during PGF and that they are not overtopped by higher cloud layers. The validity of each assumption will be briefly discussed.

- Conservative scattering: This assumption is commonly invoked in studies that estimate cloud albedo susceptibility or change (Twomey, 1991; Platnick and Towmey, 1994; Hill et al., 2008a,b; Hill et al., 2009; Chen et al., 2011). Liquid cloud particles are generally conservative (single scattering albedo ~ 1.0) for small to moderate cloud optical depth ($\tau_c < 23.0$). As we will demonstrate, marine cumulus and stratocumulus cloud layers are nearly always below this threshold during PGF.

- Invariant asymmetry: For visible light, cloud droplet scattering asymmetry varies weakly with particle size (Kokhanovsky, 2004). For liquid drops, the variation is primarily by approximately 5% over the range of effective radius from 6 µm to 19 µm. We will demonstrate that the estimated change in liquid drop effective radius during and near PGF periods lies well within this range.

- Constant liquid water path: This is the least likely of the above listed assumptions to be valid. Cloud albedo is susceptible to changes in both cloud droplet number concentration and cloud liquid water path. The latter can also vary with cloud droplet number concentration through cloud dynamics pathways including the so-called "evaporation-entrainment" and "sedimentation-entrainment" effects (Lu et al., 2005; Wood et al., 2007; Hill et al., 2009; Chen et al., 2011). The impact of these feedbacks to cloud albedo through the dynamics that control cloud liquid water path vary strongly with environmental conditions, and in some cases can cancel the direct increase in cloud albedo resulting from an increase in cloud droplet number concentration. Environmental conditions during PGF (greater likelihood of a dry free atmosphere, and an increase in large-scale subsidence and thus increased low-level static stability) have been found to favor competing effects on susceptibility through the entrainment effects (e.g. Wood et al., 2007; Chen et al., 2011) The strength of the entrainment feedbacks is strongly dependent on sea surface temperature as well. PGF can occur under a wide range of sea-surface temperatures arising from natural variability in the northeastern Pacific Ocean. To disentangle the total susceptibility which may arise from these competing liquid water path feedbacks, a series of large-scale eddy simulations, similar to those in Lu et al. (2005) and Chen et al. (2011), are required. This is beyond the scope of the current study, thus we will only estimate the so-called "Twomey Effect" (or cloud albedo first aerosol indirect effect) on albedo



which corresponds to the increase in cloud albedo due to an increase in CCN concentration when liquid water path is held fixed.

The MODIS level-2 cloud products provide swath-level retrievals of liquid cloud optical depth, liquid cloud effective radius, and cloud top pressure twice daily during daylight hours from descending (Terra – 10:15 local time) and ascending (Aqua – 13:45 local time) sun-synchronous orbits. The level 2 cloud products have a nominal spatial resolution of 20 km. For this study, daytime retrievals during PGF conditions from the expanded catalogue (N06 PGF events plus Table 3 from this study) during the MODIS operational period (2002 – present) were screened to remove pixels over land or more than 75 km from the coast (offshore extent of PGF airmass found by aircraft and reported in N06) and pixels which likely did not correspond to low-level cumulus or stratocumulus. We followed the cloud type definitions (e.g. Figure 2 from Rossow and Schiffer, 1999) from the International Satellite Cloud Climatology Project (ISCCP) that rely upon thresholds of both cloud top pressure and cloud optical depth. Pixels for which no cloud information was retrieved were also discarded (no cloud present, or retrieval algorithm failed). The retrieved effective radius was also retained to judge the suitability of the invariant asymmetry assumption. The cloud albedo change reported is thus the estimate of the Twomey effect on albedo during PGF episodes when marine cumulus or stratocumulus are present with clear sky above marine low level clouds.

## 4 Results

### 4.1 Description of PGF Cases Observed During CalWater 2015

Table 3 lists all cases which fit the mPGF requirements during CalWater 2015. Hereafter, these will be referred to as PGF(1-5). Some key parameters which describe the PGF layer flow measured by the 449 MHz radar are also summarized in Table 3, along with their 67 case rank (62 cases from N06 plus 5 from CalWater 2015). Note that in all 5 cases, both mPGF and elevated CO are met for a majority of the period, however the listed start time and duration in Table 3 is for mPGF, and in some cases the duration for PGF may be shorter than that listed when the additional elevated CO constraint is enforced.

### 4.2 Airmass Properties During PGF

Figure 2 shows a box and whisker plot for the peripheral instrument data. So that all measurements fall on the same scale, each measurement has been normalized according to it's all-study mean ($\mu_{all}$) and has been plotted according to its natural logarithm. For CO, $NO_x$, and CN, the interquartile range during PGF lies entirely above (or nearly so in the case of CN) the interquartile range during CTL. The difference in likely concentration is most dramatic for $NO_x$, for which the minimum hourly concentration during PGF is nearly the median CTL concentration and the maximum CTL concentration is nearly the median PGF concentration. Median APS number concentration is not preferentially higher during PGF, CTL or ALL hourly periods, though the range of APS for each period varies slightly. $PM_{2.5}$ is more likely to be elevated during PGF, but its interquartile range overlaps with the interquartile range of $PM_{2.5}$ during CTL and ALL hourly periods. Mean PM2.5 during CTL periods is estimated to be $14.0\pm11.9$ µg m$^{-3}$. This estimate increases to $22.9\pm16.0$ µg m$^{-3}$ during PGF



Figure 2 also displays wind rose diagrams for ALL and CTL periods. The distribution of wind directions and speeds during CTL suggest that these periods are dominated by the land-sea breeze diurnal cycle (BML is situated just east of a shoreline oriented NNW to SSE). The wind rose for PGF is not shown, since wind direction was used in the algorithm for defining PGF.

The range of normalized hourly $\sigma_{ATN}(\lambda)$ values measured during PGF, CTL and LOCAL periods are shown in Figure 3. The normalization method follows that used in Fig. 2. As discussed in the methods section, absolute values are uncorrected and thus not reported. The median and upper/lower quartile values of $\sigma_{ATN}$ were compared to published work (Table 1, Chung et al., 2012) and it was found that they are reasonable for the location and concentration/type of aerosols measured. It should be noted that absorption coefficient median and interquartile

ranges are highest during PGF, followed by LOCAL and CTL and that highest maximum values of $\sigma_{ATN}$ are observed during LOCAL. The AAE derived from the visible light channels for each period is reported in the figure annotation. The AAE (0.98 +/- 0.21) during PGF is very close to 1.0, which is widely accepted to be indicative of fresh soot (Chung et al., 2012). Fig. 3 shows that AAE during CTL (0.87 +/- 0.10) decreases during PGF. AAE is expected to decrease with soot particle age (accumulation of organic and nitrate on the particle surfaces). A possible

explanation for the decrease in CTL AAE wrt. PGF is that during PGF direct lower tropospheric transport through the Petaluma Gap brings CV and SFBA brings more freshly emitted soot particles to the measurement site. During CTL periods, fewer soot particles are present (lower absorption coefficient), and those that are measured by the aethalometer have been further aged. In section 4.4, this conclusion is supported by single particle mixing state analysis which shows that organic to elemental carbon ratio decreases during PGF periods compared to CTL. The

AAE during LOCAL periods is highest at 1.17 +/- 0.11. This value is closer to that reported for biomass burning (Clarke et al., 2007; Lewis et al., 2008) than is the AAE during PGF or CTL.

Figure 4 shows the average merged size distributions for PGF and CTL sampling periods. CTL periods were marked by lower particle concentrations in the submicron mode and higher in the coarse mode ($D_p > 1.0$ μm). CTL periods often experienced westerly flows and would be expected to be dominated by marine aerosol from the

Pacific Ocean. The marine boundary layer over the remote ocean is typified by low particle concentrations and a significant supermicron mode (Quinn et al., 2015). Integrated average supermicron counts on the APS during CTL periods were 14.9 cm$^{-3}$, compared to 2.4 cm$^{-3}$ for PGF periods, a decrease of 84%. The Fig. 4b is a log$_{10}$-log$_{10}$ plot of $D_p$, and shows supermicron particles concentrations approximately an order of magnitude greater. Single particle composition data from the ATOFMS during these time periods, discussed more in section 4.3, confirmed that the

increase in coarse mode particles could be attributed to greater concentrations of marine-type particles.

PGF events, in contrast, showed a large increase in the number of particles with $D_p < 1.0$ μm, and a new ultrafine mode with mode diameter $D_p = 36$ nm. The Fig. 4a shows this relationship more clearly, and the corresponding decrease in coarse mode particles. CN increased by 110% during PGF compared to CTL, from 311.0 cm$^{-3}$ to 650.9 cm$^{-3}$, These results were also expected, because continental, anthropogenically-influenced

airmasses typically contain smaller and more numerous particles (Tunved et al., 2005).This increase in particle number during PGF events was correlated with the increase in PM$_{2.5}$ as shown in Fig. 2.

**4.3 Aerosol Particle Types**




PGF conditions coincide with a shift in particle type away from marine and towards continental. Figure 5 shows pie charts of the sub- and super-micron particle populations for CTL vs PGF. Percentages indicate the number fraction of particles assigned to the corresponding particle type. The total hit rate for all particles were 20.8%. Panels (a) and (b) show the submicron (0.2-1.0µm) particle number fraction by type during CTL (1,222,274 particles) and PGF (164,952 particles), respectively. PGF had approximately the same level of submicron biomass burning particles (BB). However, there was a large increase in elemental carbon/organic carbon (ECOC) (20-28%) and ammonium nitrate (AN) (10-28%) particle types. These two types have both been linked to anthropogenic activity. The ECOC type has been observed before with the aging of fuel emissions (Hughes 2000) in the Los Angeles basin during stagnant conditions with high pollutant concentrations. Similarly, the California Central Valley is an area of elevated hydrocarbon fuel emissions and frequent long-lived lower tropospheric inversions, and might be expected to support the formation of ECOC particles. AN particles have been linked to the accumulation of ammonia and nitric acid on particles (Qin et al., 2012) and nucleation by reactions between ammonia and nitric acid (Russell and Cass, 1986; Hughes et al., 2002), gases strongly correlated with anthropogenic activity. Marine aerosols make up a sizable fration of submicron particles during CTL (SSA and aged SSA). However, these particles are much less prominent during PGF. The increase in ECOC and AN and decrease in marine particle types reinforce the conclusion that PGF airmasses originate continentally and have strong anthropogenic character.

The clearest delineation in particle type was observed in the supermicron fraction (1.0-3.0 µm). Panels (c) and (d) of Fig. 5 show that over 90% of CTL supermicron particles (544,612 total particles) were either fresh or aged marine particles, while less than 20% were of marine origin during PGF. During PGF, supermicron particles (25,457 total particles) were comprised of primarily BB, ECOC, AN and elemental carbon (EC), a byproduct of fuel combustion. Additionally, the majority of PGF marine particles showed markers of reacting with nitric acid (Gard et al., 1998). This contrasts CTL marine particles, which were primarily unreacted.

The Dust and Dust/Bio types also increased during PGF. The CV, despite its agricultural production, is a semi-arid environment and can be a significant source of dust. Conversely, BML airmasses during CTL periods were heavily influenced by the Pacific Ocean and thus were not expected to contain much dust. The shift in supermicron particles composition away from marine particles to anthropogenic and dust particle types supports the conclusion that PGF airmasses likely originate from the CV.

### 4.4 Aging Processes Observed Through Secondary Species Markers

Figure 6 shows the sulfate:nitrate ratio (SN) of particles separated by type. For visual clarity, the SN ratio has been normalized so that ratios <1 will approach -1 as it proceeds to -∞ (i.e. nitrate without sulfate) and a ratio >1 will approach 1 as the ratio proceeds to +∞ (i.e. sulfate without nitrate). As ratios approach −1 or 1, they are exponentially increasing, while close to zero, the RPA of each species is essentially the same. This results in a broader range of ratios for bins near −1 or 1, while the bins near 0 bins include a smaller range of ratios. The left panel depicts the SN for CTL particles. The majority of particles showed higher SN ratios, indicating that aging primarily occurred through the accumulation of sulfate. The exception to this rule was the SSA type, which can



react with $NO_x$ species in a displacement reaction to liberate HCl (Gard et al., 1998; Cahill et al., 2012). PGF particles showed a SN ratio biased toward nitrate accumulation, indicating that the primary and most important aging mechanism was through $NO_x$ pathways.

In addition to probing the partitioning of acidic species, basic species like amines and ammonium were investigated. Figure 7 shows the normalized ratio for amines:ammonium ratio (AA). The AA for CTL particles shows fairly equal partitioning for all particle types. During PGF the average peak area of amine peaks ($^{58}C_2H_5NHCH_2^+$, $^{59}NC_3H_9^+$, $^{86}(C_2H_5)_2NCH_2^+$) in particles actually increased, but the AA ratio shifted toward ammonium because the ammonium content of particles increased much more. The area surrounding BML contains animal husbandry, but no industrial-scale farming. The shift in basic species partitioning indicates that PGF

particles originate from a large source of ammonium. The CV region contains many more industrial-scale farms where ammonium is employed, and thus this change in aging shows that the PGF airmass likely originates within the CV.

Previous studies (Cahill et al., 2012) have used the ATOFMS to determine the internal mixing state of carbonaceous particles. Figure 8 shows the organic carbon:soot ratio as calculated by the ATOFMS, seperated by

particle type. CTL particles appear to have relatively higher amounts of OC, most notably in the Amine particle types and, unsurprisingly, the OC type. The ratio plot indicates that PGF particles contain more EC relative to CTL particles. This is despite the appearance of the AN particle type, which had greater OC character. These OC/EC ratio plots agree with the aethalometer-derived AAE, which suggest that the soot was less aged during PGF than during CTL periods.

In summary this analysis shows that the preeminent aging mechanisms associated with PGF are the accumulation of ammonium and nitrate, in accordance with previous studies on Central Valley particle composition (Qin et al., 2006). Amine accumulation was also observed in ECOC and AN particless, but was determined to not be as significant as ammonium. Accumulation of nitrogen species on aerosol particles is important as it increases the risk of nitrogen deposition into coastal waters, which can lead to ecosystem

degradation (Ryther and Dunstan, 1971; Paerl, 1995; Paerl, 1997). The shift toward internal mixtures containing elemental carbon and away from particulate matter containing primarily organic carbonaceous species during PGF suggests that gap flow may cause increased solar absorption by near surface aerosols, especially in visible wavelengths. This potential impact is corroborated by the aethalometer PGF and CTL measurements.

**4.5 CCN and Cloud Droplet Spectra**

Figure 9 displays the cumulative CCN supersaturation spectrum (versus liquid supersaturation) transformed from the size-resolved CCN data and the cloud droplet number concentration (CDNC) for updraft velocities between 0.1 m s$^{-1}$ and 10 m s$^{-1}$. CCN concentration is enhanced during PGF by 2.8 to 3.0 for a wide range of supersaturations. The increase in CCN is remarkably consistent across the range due to the confluence of two factors. First, the hygroscopicity parameter – or the contribution of particle chemistry to CCN activation – does

not change significantly between PGF and CTL (0.21 vs. 0.20). Second, particle concentrations are larger for all sizes D < 500 nm during PGF events (Fig. 4). These sizes dominate the spectra in Fig. 9a. As a consequence CDNC



increases across all considered cloud updraft speeds during PGF episodes. Figure 9b shows that CDNC increases between 125% and 145%, and that this relative increase is expected for all possible cloud types. In the results to follow concerning cloud albedo change during PGF, CDNC increases are considered as a ratio (i.e. $\chi = N_{PGF}/N_{CTL}$). In this framework, The average ratio, $\chi$, is 2.35.

### 4.6    Impact of PGF on Marine Cumulus and Stratocumulus Albedo

Figure 10 shows the range of both MODIS estimated cloud albedo and the Twomey effect albedo change that results from the modeled increase in CNDC during PGF compared to CTL periods. As discussed in the methods section, the values reported here correspond to PGF episodes where marine cumulus or stratocumulus are present within 75 km of the shoreline and when clear the marine cloud layer is topped by clear sky. During expanded catalog PGF episodes that match these conditions, the observed cloud albedo ranged from 0.01 to 0.63, with upper (lower) quartile values of 0.43 (0.17). Using the observed $A_c$ = 0.43 (0.17) and $\chi \sim 2.35$, equation 7 from Platnick and Twomey (1994) estimates that $\Delta A_C/A_C = 0.28$ (0.16). Therefore, clouds that entrain the PGF aerosol without concomitant changes in liquid water path are expected to be 16% to 28% more reflective.

### 5    Summary

Measurements taken at Bodega Bay, CA during the CalWater 2015 intensive observing period were used to investigate the impacts of Petaluma Gap Flow on local air quality and marine cloud albedo. The kinematics of PGF and its relation to synoptic scale weather patterns and the Central Valley cold pool has been perviously described in N06. This study is the first attempt to quantify the impact of PGF on the boundary layer airmass and particle chemistry.

Vertically resolved lower tropospheric wind observations and carbon monoxide concentration were used to identify PGF periods during the CalWater 2015 intensive observing period and separate these from CTL periods, during which the BML airmass was neither influenced by PGF or heavy pollutant loads from a local source. Five PGF events were identified during Calwater 2015 and were compared to the PGF catalog published in N06 by means of their local weather attributes.

During Calwater 2015 PGF periods, several measures of anthropogenic pollution, including CO, $NO_x$ CN, and black carbon mass concentration estimated by a multi-channel Aethamometer, were consistently elevated when compared to CTL periods. Using SMPS and APS aerosol size spectrometers, we found that aerosol number concentrations increase by 110% in the submicron size range, while decreasing 84% in the supermicron size range. Both PGF and CTL periods presented size distributions with a similar accumulation mode near 100 nm. The PGF period composite size distribution contained a prominent mode below 50 nm which was not present in the CTL composite. This fine mode indicates that particle source and/or degree of particle chemical aging may be significantly different during PGF periods. The particle chemistry of this fine mode could not be investigated because the relevant size are below  the lower detection limit of the UF-ATOFMS PGF periods contained 84% fewer coarse mode particles than did CTL periods. The relative lack of particles at these sizes is related to a significant change in super-micron particle chemistry found by analyzing single particle mass spectra. Taken together, the above results demonstrate the change in aerosol physicochemical properties during PGF events.





Single particle chemical mixing state during PGF events was investigated using UF-ATOFMS and ATOFMS Measurements. It was found that submicron particle populations change during PGF to favor ECOC, BB, AN, and EC types at the expense of SSA types. The large difference in supermicronparticle mixing state is likely related to the shift in prominent wind direction during PGF. The analysis of secondary aging also showed

that carbonaceous particles are more likely to contain elemental carbon than organic carbon during PGF episodes. Aethalometer-derived AAE also suggested that observed soot was less aged during PGF periods but total absorption and total black carbon mass were greater than during CTL. The above results reinforce the hypothesis that PGF could lead to an increase in absorption of solar shortwave radiation by black carbon aerosol, which may be associated with more freshly emitted soot.

PGF and CTL single particle mass spectra relative peak area ratios were used to investigate particle aging mechanism. PGF particles were much more likely to acquire nitrate than CTL particles, which preferentially contained sulfate. This was especially true for AN, ECOC, BB, and EC particle types during PGF, but may not appply to SSA and Aged SSA, which showed a preference for nitrate aging even during CTL periods. The aging of SSA by nitrate is a well-documented phenomenon that was also regularly observed during CalWater 2015.

Relative peak area analysis also showed that particles are much more likely to chemically age by ammonium than by amines during PGF. This tendency appeared especially strong for BB, EC and ECOC types. While OC type particles increased in relative number during PGF episodes, they appeared to favor the amine aging pathway even during PGF. Together the above results reinforce the hypothesis that PGF could lead to increased deposition of nitrogen containing particulate matter to the local ecosystem near and offshore of Bodega Bay. This result may

also be true in other coastal locations which are periodically influenced by offshore gap flow originating in a $NO_x$ and ammonia enriched airmass (e.g. the nearby Salinas Valley, and offshore of the Golden Gate). If increased nitrogen deposition is occurring during PGF episodes, it could lead to eutrophication and algal blooms, as suggested by Paerl (1995; 1997).

Particle hygroscopicity, as shown by size-resolved CCNc measurements, was nearly invariant between

PGF and CTL periods. An adiabatic parcel model was usedto estimate the cloud droplet number concentration resulting from the derived CCN activation curves. The increase found in CDNC was stable across a wide range of updraft velocities. . The marine cloud albedo change in response to PGF CCN was estimated using MODIS level 2 cloud products and equation 7 from Platnick and Twomey (1994). To first order (assuming constant liquid water path) it is estimated that near shore marine clouds will brighten by 16% to 28% percent (interquartile range) in

visible wavelengths during PGF events. This finding supports the hypothesis that PGF conditions may lead to a brightnening in near-shore marine stratocumulus clouds through the cloud albedo indirect effect.

The conclusions reached in addressing the three hypotheses posed in this study represent only the first attempt to characterize the impact of Petaluma Gap Flow on the aerosol direct effect, aerosol indirect effect and coastal environment in Northern Central California. Due to the relatively short CalWater-2 intensive observing

campaign, these results were drawn from only 5 PGF events. The data necesary to investigate these hypotheses was drawn from a large multi-agency effort including many specialized and operator intensive measurements, which by nature must be short in duration. Longer term observation, perhaps by less detailed but targeted chemical observations at similar locations could significantly augment the findings presented here.





During this study, we attempted to detect inter-event differences in relative peak area ratios for secondary aging indicators, but no significant change was observed. In addition, the authors wish to comment that many of the assumptions made (e.g. constant liquid water path) in estimating the impact of PGF on marine cloud albedo change can only be discarded through airborne observations or modeling studies. These were considered beyond the scope of this study, but may be valuable future investigations to fully describe the impact of polluted offshore-directed gap flow on marine cloud brightness.

The findings presented herein demonstrate that PGF can impact aerosol number, chemical aging pathways, shortwave absorption and the number of CCN available to near-shore marine clouds. These findings are the first of their kind that result from direct observation of an intermittent weather phenomenon that brings anthropogenic pollutants to an otherwise remote region. The authors argue that further study of the chemical composition of continental outflow in other regions is necessary to refine current understanding of the impact of human activities on the environment.

### Acknowledgements

The authors thank all other CalWater and ACAPEX 2015 participants including those from Pacific Northwest National Laboratories, The National Oceanic and Atmospheric Administration, NASA Jet Propulsion Laboratory, the Naval Research Laboratory, University of California, Davis, Scripps Institution of Oceanography, Colorado State University and North Carolina State University. The authors would also like to thank the UC Davis Bodega Marine Laboratory for the use of laboratory and office space and shipping and physical plant support while collecting data, as well as the California Air Resources Board and the National Park Service for the trailers used for sampling. This research was funded by NSF award number 145147 (ACM, GCC, KAM, KAP), NSF award number 1450690 (MDP, NR, HT), and NSF award number 1450760 (SAA, SMK, PJD)



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





**Table 1. Decision tree used for filtering measurement periods, and the resulting number of hourly periods (/total) in each category.**

| Decision Criteria | | 3 PGF Criteria at BBY (Neiman et al., 2006) | |
|---|---|---|---|
| CO conc. greater than | | Y | N |
| $\mu + 2\sigma$ | Y | PGF (55/1248) | Local (407/1248) |
| (138.1 ppbv) | N | Onset/Diffuse (11/1248) | CTL (775/1248) |





**Table 2. Summary of particle types determined by ATOFMS and their characteristic ion markers.**

| Particle Type | Characteristic Peaks |
|---|---|
| Amines | $^{58}C_2H_5NHCH_2^+$, $^{86}(C_2H_5)_2NCH_2^+$ (Angelino *et al.,* 2001; Pratt *et al.,* 2009; Qin *et al.,* 2012). |
| Ammonium nitrate (AN) | $^{18}NH_4^+$, $^{30}NO^+$, $^{46}NO_2^-$, $^{62}NO_3^-$, $^{97}HSO_4^-$, $^{125}(HNO_3)NO_3^-$ (Pastor *et al.,* 2003; Qin *et al.,* 2012). |
| Biomass burning (BB) | Strong $^{39}K^+$ and $^{97}HSO_4^-$, less intense $^{12}C^+$, $^{26}CN^-$, $^{46}NO_2^-$, $^{62}NO_3^-$, $^{125}H(NO_3)_2^-$ (Silva *et al.,* 1999). |
| Elemental carbon (EC) | Carbon Clusters at $C_n^+$ and $C_n^-$ (Moffett and Prather, 2009; Spencer and Prather, 2006). |
| Elemental carbon/organic carbon (ECOC) | $^{12}C^+$, $^{24}C_2^+$, $^{27}C_3^+$, $^{36}C_3^+$, $^{37}C_3H^+$, $^{43}CH_3CO^+/CHNO^+$ (Moffet and Prather 2009; Qin et al., 2012). |
| Organic carbon | $^{27}C_2H_3^+/CHN^+$, $^{37}C_3H^+$, $^{43}CHNO^+$ (Silva and Prather, 2000; Spencer and Prather, 2006; Qin et al., 2012). |
| High mass organic carbon (HMOC) | $^{37}C_3H^+$, $^{43}CHNO^+$, differences of 14-16 past 150 m/z, $^{46}NO_2^-$, $^{62}NO_3^-$, $^{97}HSO_4^-$ (Denkenberger et al., 2007; Qin et al., 2006) |
| Dust | Inorganic ions $^{6/7}Li^+$, $^{27}Al^+$, $^{39}K^+$, $^{40}Ca^+$, $^{48/64}Ti/TiO^+$, $^{54/56}Fe^+$, $^{60}SiO_2^-$, $^{76}SiO_3^-$, $^{79}PO_3^-$ (Silva et al., 2000). |
| Dust/bio | Same as Dust, but also with biological markers $^{26}CN^-$, $^{42}CNO^-$ |
| Aged marine (Aged SSA) | Same as Fresh Marine but also with $^{108}Na_2NO_3^+$, $^{46}NO_2^-$, $^{62}NO_3^-$, $^{147}Na(NO_3)_2^-$ (Gard *et al.,* 1998). |
| Fresh marine (SSA) | $^{23}Na^+$, $^{24}Mg^+$, $^{39}K^+$, $^{40}Ca^+$, $^{81,83}Na_2Cl^+$, $^{35,37}Cl^-$, $^{58}NaCl^-$, $^{93,95,97}NaCl_2^-$, $^{151,153,155}Na_2Cl_3^-$ (Gard et al., 1998). |





Table 3. PGF events observed during CalWater 2015 and their significant parameters following N06. Ranks are out of 67 (62 cases from N06 plus 5 from CalWater 2015).

| Start date @ time (UTC) | Duration (hr) (rank) | Jet Altitude (m MSL) (rank) | Jet Maxima (m/s) (rank) | Jet Dir. (degree) | Gap Folding Height (m MSL) (rank) | Vertical Shear across folding alt (m/s) (rank) | Precipitation at BBY (mm) (rank) |
|---|---|---|---|---|---|---|---|
| 1/14/2015 @ 1300 | 31 (5) | 307 (47) | 10.0 (16) | 96 | 622 (20) | 12.8 (46) | 0 (67) |
| 1/25/2015 @ 1100 | 8 (44) | 1146 (1) | 15.7 (2) | 79 | 1776 (1) | 3.7 (67) | 0 (67) |
| 1/26/2015 @ 1200 | 10 (33) | 202 (47) | 8.6 (37) | 98 | 517 (28) | 12.4 (48) | 0 (67) |
| 2/04/2015 @ 0700 | 9 (38) | 202 (47) | 8.8 (33) | 122 | 517 (28) | 7.8 (62) | 0 (67) |
| 3/05/2015 @ 1300 | 8 (44) | 412 (6) | 7.6 (56) | 119 | 727 (12) | 11.6 (50) | 0 (67) |





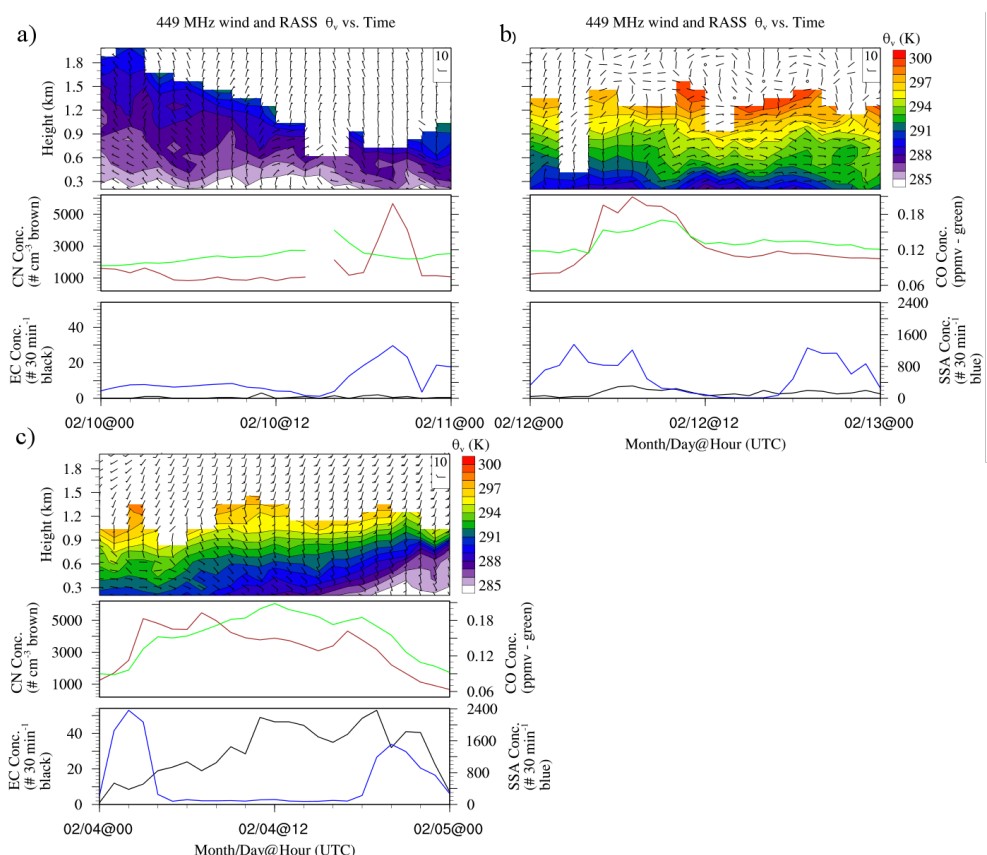

**Figure 1: a) Horizontal wind barb and $\theta_v$ every 100 meters from 300 m AGL to 1900 m AGL from NOAA-ESRL 449 MHz wind prifiling radar and RASS at BBY (top), hourly CO concentration (ppmv – green) and CN (# cm$^{-3}$ – brown) (middle), Number particles classified as EC or Aged EC (black) and SSA or Aged SSA (blue) per 30 minute interval from ATOFMS (bottom) during a 24 hour period (02/10/2015) calssified as CTL. b) as in a, except for a 24 hour period (02/12/2015) classified as LOCAL. c) as in a, except for the 24 hours surrounding PGF 4.**





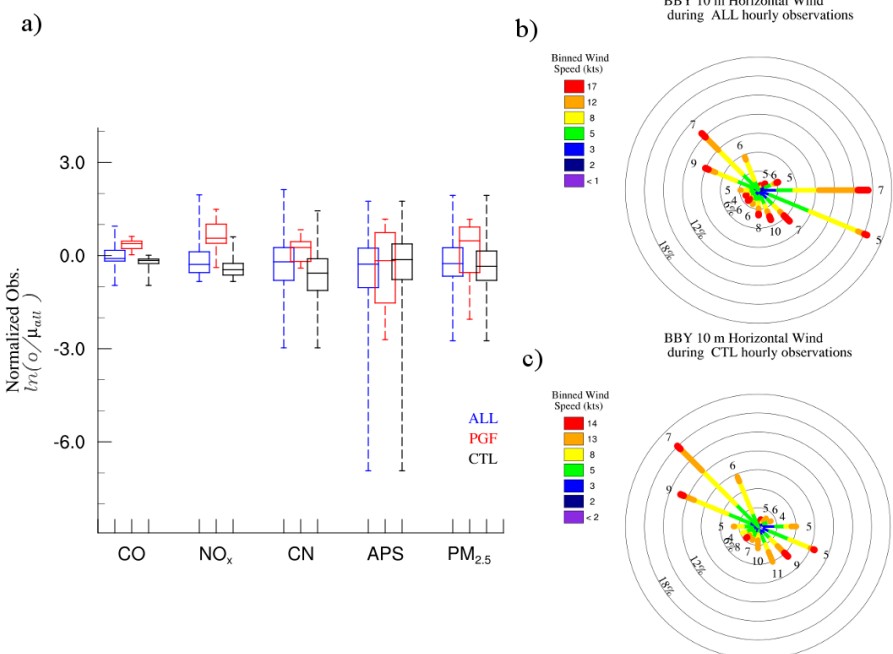

**Figure 2: a) Box and Whisker Plot displaying normalized peripheral measurements during all (ALL – blue) hourly CalWater 2015 periods, PGF (PGF – red) periods, and CTL (CTL – black) periods. b) BBY 10 m wind rose diagram for ALL. Rings represent probability of wind from displayed direction, colors represent relative probability of wind speed exceeding displayed threshold, numbers at end of petals represent mean wind speed from displayed direction. c) As in b, except for CTL periods.**



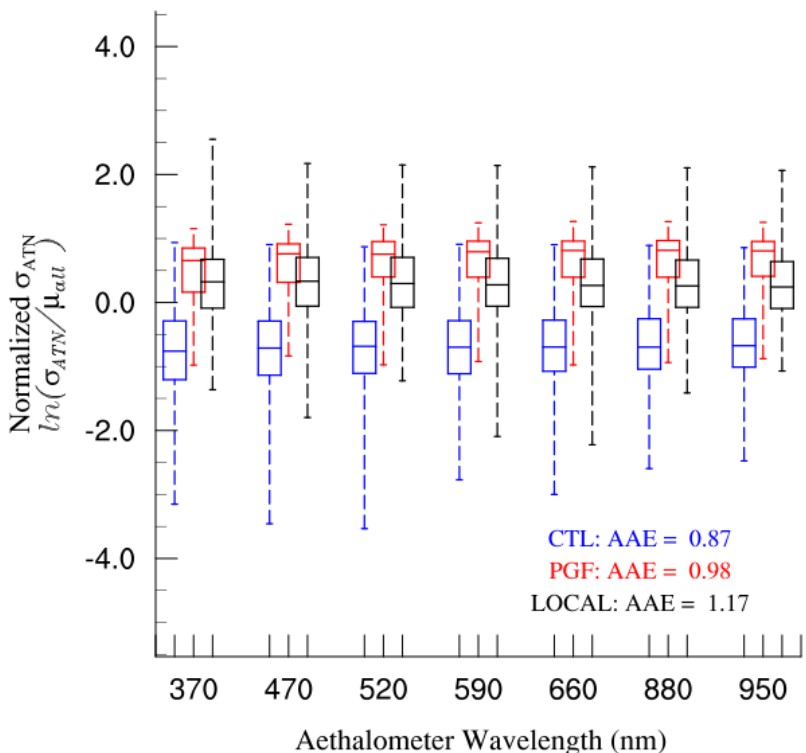

**Figure 3: Normalized Aethalometer Light Absorption Coefficient at seven wavelengths for hourly periods classified as CTL (blue), PGF (red) and LOCAL (black). Upper/Lower box bounds represent upper/lower 25% values, respectively. Upper/Lower whiskers represent max/min values respectively. Box middle represents median value. Also displayed are the AAE values found by regression during each period.**





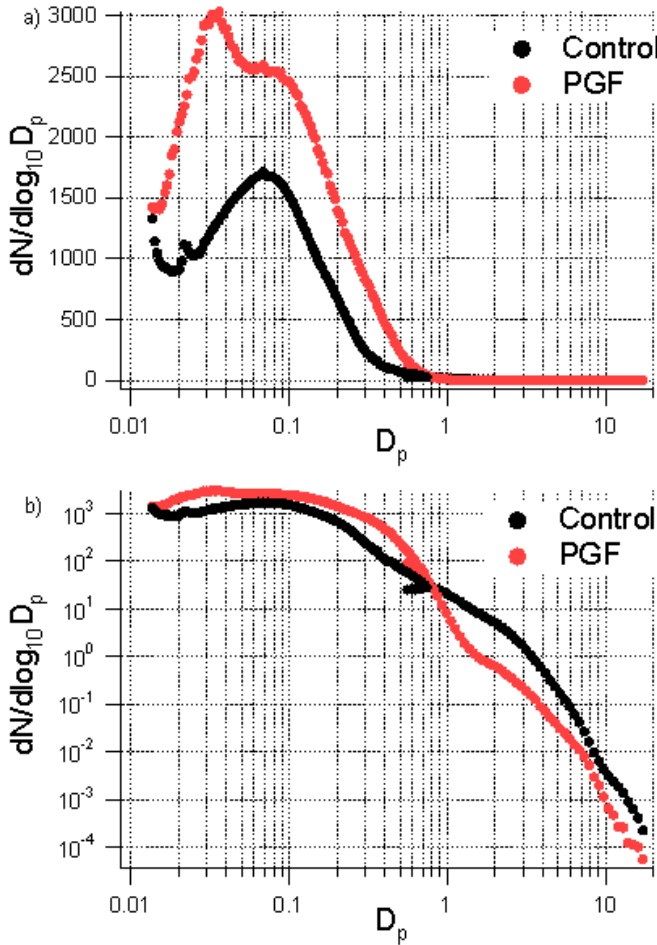

**Figure 4: a) Composite merged SMPS-APS size distribution displayed as $dN/dlog10D_p$ for PGF (red) and CTL (black) periods. b) as in a, except displayed on a log-axis.**



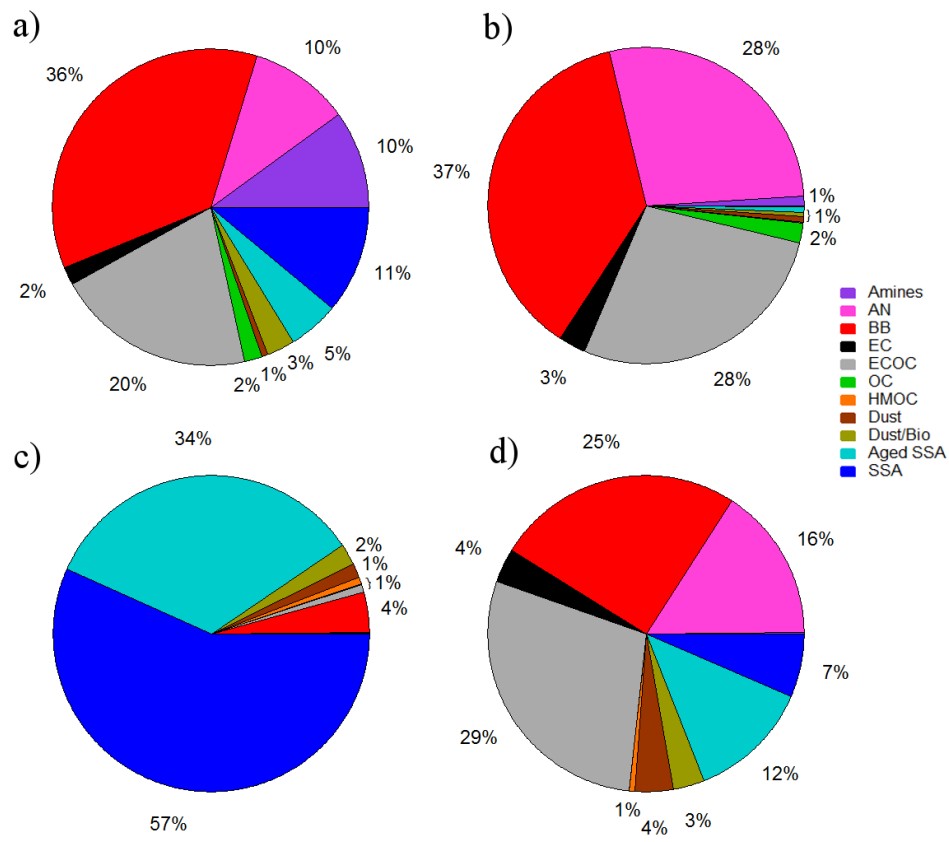

**Figure 5: Pie charts for sub- (top panels) and supermicron (bottom panels) particle types for CTL (left panels) and PGF (right panels).**





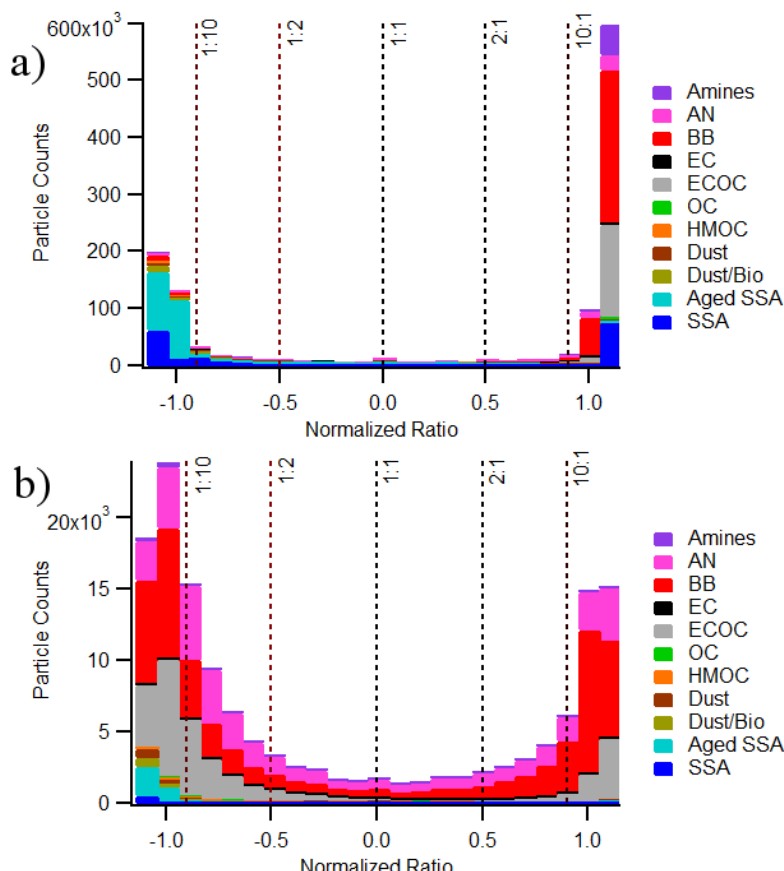

**Figure 6: a) Particle sulfate:nitrate ion ratio distribution for CTL periods. Values <0 indicate more nitrate than sulfate and values>0 indicate more sulfate than nitrate. Ratios representing 1:1, 2:1, and 10:1 are shown by vertical dashed lines. Significant particle types are represented by separate colors. b) as in a), except during PGF periods.**





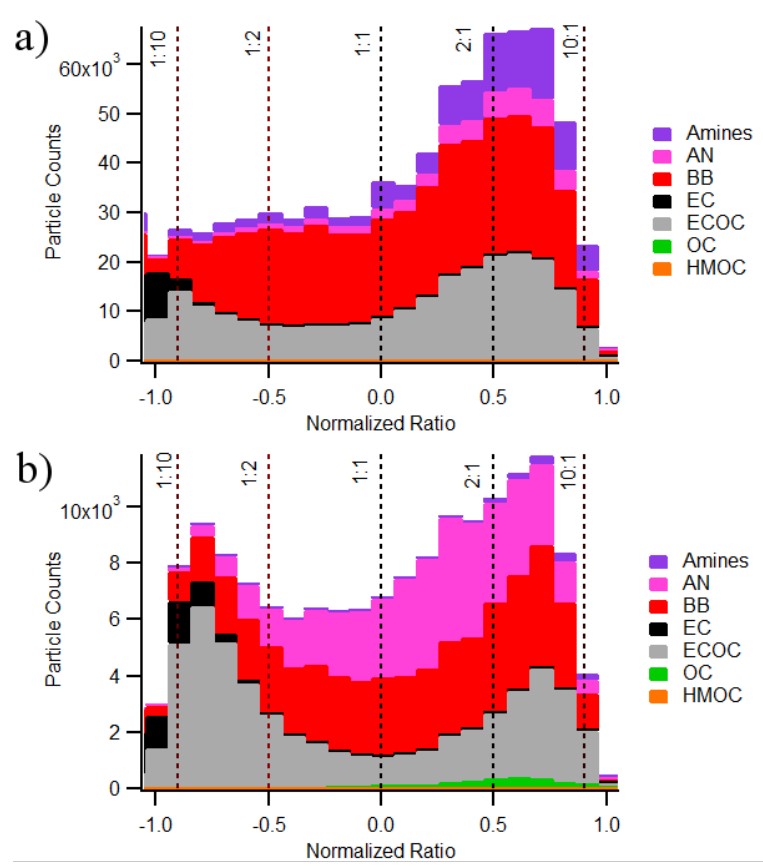

Figure 7: As in Figure 6, except amines:ammonium ion ratio distributions are shown.





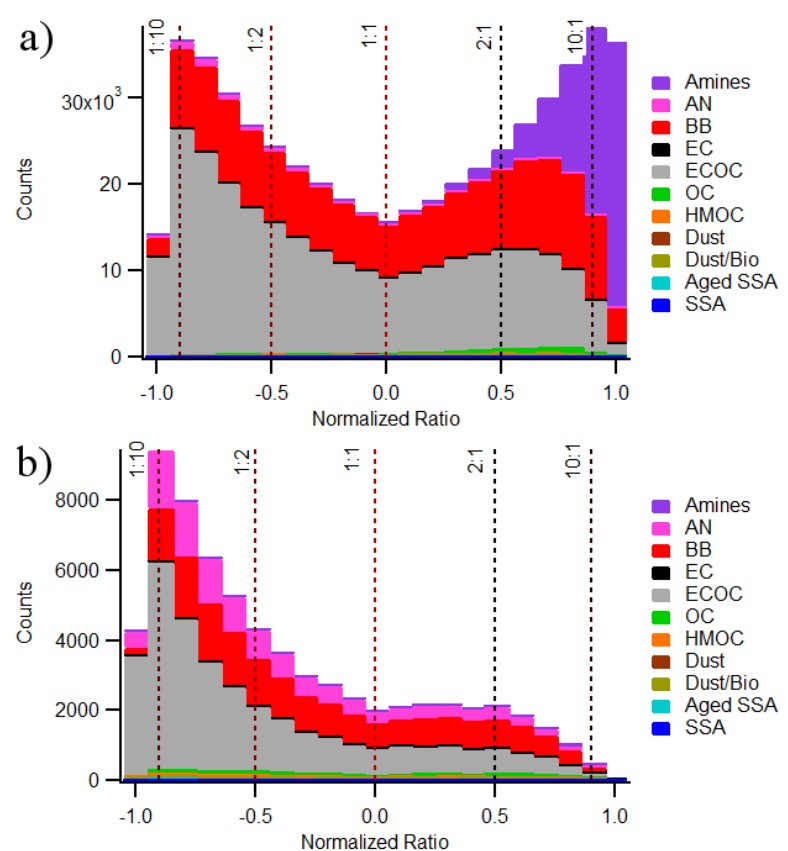

**Figure 8: As in Figure 6, except OC:soot ion ratio distributions are shown.**





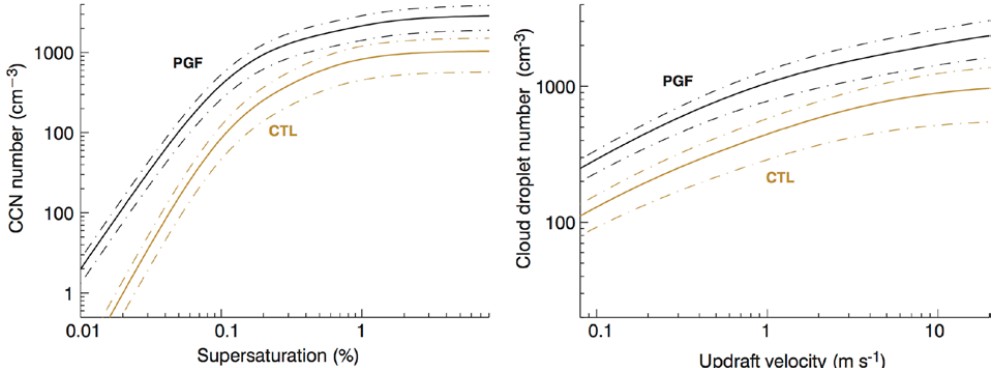

**Figure 9: Left) cumulative median CCN supersaturation spectrum PGF periods (black) and CTL (yellow). Dashed lines approximate the interquartile range. Right) as in left, except for predicted cloud droplet number concentration as a function of updraft velocity.**




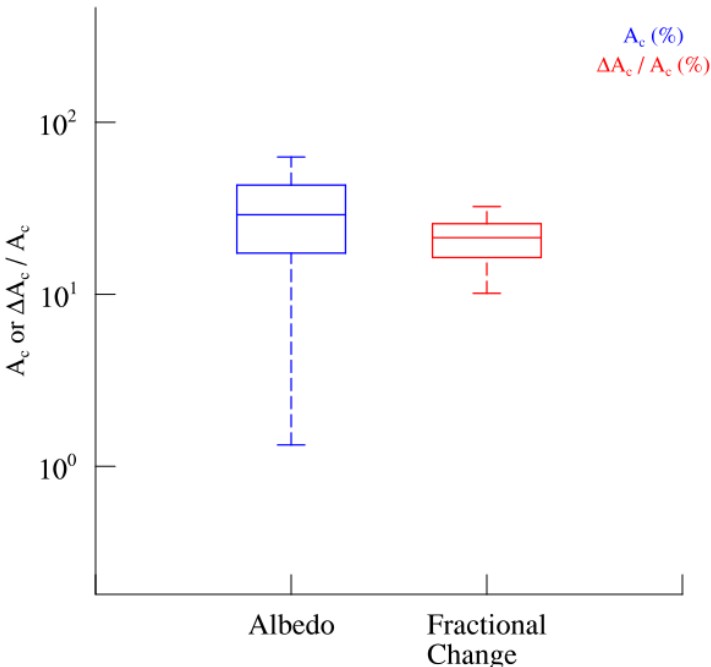

**Figure 10: Left) Box-and-whisker diagram representing the shortwave (0.64 μm) albedo of near-shore marine cumulus and stratocumulus clouds detected by MODIS during expanded catalog PGF episodes. Right) As in left, except for fractional change in marine cumulus and stratocumulus albedo.**

