# Peer review of "Transport of Pollution to a Remote Coastal Site during Gap Flow from California's Interior: Impacts on Aerosol Composition, Clouds and Radiative Balance"

_Atmospheric Chemistry and Physics, 2016_

## Referee Comment (RC1) · Anonymous Referee #1 · 5 Jul 2016

Review of "Transport of Pollution to a Remote Coastal Site during Gap Flow from California's Interior: Impacts on Aerosol Composition, Clouds and Radiative Balance" by Martin et al.

This study examines ground based in-situ aerosol properties at a remote location in California and shows that the atmospheric aerosol, and with it also cloud properties, at this location depend on the atmospheric flow patterns. Anthropogenically polluted air massed can be advected under certain circumstances, but can also originate from areas closer by. The study distinguishes between different phases with different aerosol

properties, based on different air mass origins. Further, some modelling efforts are done to estimate the influence of the differences in the aerosol on clouds. Overall, the study gives a good overview concerning the conditions at this location and shows that there are possible influences of the aerosol on clouds on a regional scale. It is up to date and well written. I only have minor issues which I mention below.

As a correlation of air masses to their origin seems to be of particular importance in this work, I wonder why trajectories were not used at all. Also, I wonder if the authors could comment on the fact if their observation are particularly important to only the location of the study, or to sea side sites in general, or even to more locations worldwide. This could be discussed in a few sentences in the summary. It would also be nice if they related their results to the points they raise in the introduction (bulleted point list) in the summary, to see how their results match with the here cited literature.

The issues I raise here are, however, only thought as additional ideas the authors may pick up. Below a number of minor changes are suggested, and after they will have been addressed, together with considering what I wrote above, the manuscript is fit for publication in ACP.

Requested minor changes:

page 3, line 12: Maybe include a map of the area showing the mountain range (at least some of it), the gap and the sampling location.

page 3, line 24: Say explicitly that some more information on the instrumentation follows below. And mention explicitly when the measurements took place (which month, and for how long).

page 4, line 19: "PSL" was already introduced above (line 10 on this page), but as PSLs - homogenize and use acronyms once you have defined them.

page 4, line 19: Add values for the RH that were generally observed, and the maximum values.

page 4, line 25: This section, short as it is, is not only "Aerosol Mass". More correct would be "Aerosol and BC mass concentration".

page 5, line 2: Check the symbol in here - I guess it should have been a "kappa", but in my version, it was a very strange symbol instead.

page 5, line 27: A sentence or two describing the methodology of N06 would be good, as this is the basis for a crucial part of your work.

page 5, line 34-35: I was puzzled about the use of mPFG and PFG. Maybe explain the difference between the two explicitly in a sentence where mPFG first appears. And make sure you use both consistently in the text.

page 7, line 2: Refer to section 3.1 (otherwise it is a self-reference).

page 8, line 28: What do you mean by a "dry free atmosphere"?

page 9, line 25: Some formatting error - too large spaces between words at the end of the line.

page 9, line 28: What do you mean by "the difference in likely concentration"? Reformulate.

page 10, line 31: The size distributions shown in Fig. 4: Are they averages for all times of CTL and PGF, or just single distributions you used as examples? Explain in the text or caption.

page 13, top: In the text (related to Fig. 9b), it needs to be explained how CDNC were obtained. (It is not enough to mention an adiabatic parcel model in the summary.)

page 13, line 6 ff: Mention explicitly that the "Twomey effect albedo change" is what is shown in Fig. 10 as "fractional albedo change". I understand that in the text you try to relate the albedo measured by MODIS to the derived fractional albedo change, but I find this part of the text rather confusing. Consider rewriting this part.

page 13, line 12: "entrain" is not the best choice of word (a cloud is not a thing that is moved from an air mass with CTL properties into an air mass with PGF properties), it is rather that the clouds form in the air mass, which has a certain aerosol in it. Reformulate.

page 14, line 13: When you first mention aged SSA, "aged" is not capitalized, but it is capitalized here an in other locations. Be coherent.

page 14, line 27: Remove the extra ".".

Fig. 2 and 3: The change in color code is confusing (blue is "all" in Fig. 2, but CTL in Fig. 3, red is PGF in both, and black is CTL in Fig. 2 but local in Fig. 3). Please change this, and keep the colors for CTL and PGF for all figures where possible and applicable (e.g., Fig 4 and Fig. 9).

Fig. 5: Explain that the explanation of the abbreviations can be found in Tab. 2.

---

## Referee Comment (RC2) · Anonymous Referee #2 · 11 Jul 2016

The manuscript by Martin et al. describes the transport and secondary aging of particulate pollutants observed at a remote coastal site during channeled flow through a prominent gap from the California's Central Valley. The Petaluma Gap Flow (PGF) events were identified from local pollution and background conditions using meteorological and gas-phase measurements. Physiochemical properties of aerosol particles are reported for different airmass origins. Further, implications on aerosol indirect effects are discussed based on measurements and modeling work. In general, I found this study was conducted carefully and the paper is informative. I would recommend publication in ACP once the authors address the specific comments below.

[Figure]

Specific comments

1. Although the authors have provided a brief introduction for the phenomenon of PGF and also referenced in a previous paper (Neiman et al., 2006), it would be nice if they could also provide a map. The map may show the terrain of the Northern California region (and the prominent gap), the site location, and a diagram of the PGF to help the readers to understand the PGF without reading the reference.

2. This paper used many acronyms and abbreviations. I would suggest including a table with abbreviations for efficient reading.

3. Related to the comment above, please choose a different abbreviation for marine aerosol than the "SSA" (sea salt aerosol?), because "SSA" can also represent Single-Scattering Albedo in the aerosol and cloud community.

4. Section 2.3: Specify the size range for the APS. What are the typical RH values in the sampling line? Does the RH sufficiently low such that the size measurements were not affected?

5. Section 2.5: I was confused about whether the CCN measurements were conducted for super-saturation scan or diameter scan. Please specify.

6. Section 3.2, Page 6 Line 25: Say explicitly what correction is needed for the aethalometer. Correction for back scattering?

7. Section 4.2: Report the "all-study mean" values, and the percentage differences of mean values between the PGF and CTL periods.

8. Fig. 2: Do the colors in the wind rose plots represent the binned wind speed, as stated in the legend, or a relative probability, as stated in the caption? Also, use SI unit m s-1 instead of kts?

9. Fig.2 vs. Fig. 4: Fig. 4 shows that the mean value of APS particle number concentration was about one order of magnitude lower during the PGF than that during the

CTL (2.4 cm-3 vs 14.9 cm-3, as mentioned in Page 10, Line 27). However, their median values were very similar, as shown in Fig. 2a. Please provide some explanation.

10. Section 4.4: I found the normalization of ratios not intuitive. Add an equation in the main text, or in an appendix.

11. Section 4.4: Any explanation for the association between high OC:soot ratio and high abundance of amines-type particles (Fig. 8a)? Do the OC ions usually nitrogen containing for the CLT cases?

12. Fig. 10: I found this figure is confusing and less informative. As the values have been mentioned in the text, this figure can be eliminated. The corresponding method section (section 3.5) could be significantly shortened.

Technical comments

1. Page 5, Line 2: Please check the symbol (kappa?).

2. Page 7, Line 20: missing "." after "radiative effects"

3. Page 7 Line 24: Extra "." after "particle aging"
* * *

---

## Author Comment (AC1) · 5 Sep 2016

We would like to thank the reviewer for their valuable comments. We have reworked the paper to address the relevant issues where necessary. The reviewer comments are written in plain text, our response and changes to the manuscript in bold.

*Specific comments:*
1. Although the authors have provided a brief introduction for the phenomenon of PGF and also referenced in a previous paper (Neiman et al., 2006), it would be nice if they could also provide a map. The map may show the terrain of the Northern California region (and the prominent gap), the site location, and a diagram of the PGF to help the readers to understand the PGF without reading the reference.

**Added new figure 1, which is a map of the study site, relevant geographical features in CA and the Petaluma Gap Flow.**

[Figure]

2. This paper used many acronyms and abbreviations. I would suggest including a table with abbreviations for efficient reading.

**Included an additional table (Table 4) to with a list of acronyms and abbreviations.**

3. Related to the comment above, please choose a different abbreviation for marine aerosol than the "SSA" (sea salt aerosol?), because "SSA" can also represent Single Scattering Albedo in the aerosol and cloud community.

**Changed all references of SSA to SS in text and in figures.**

4. Section 2.3: Specify the size range for the APS. What are the typical RH values in the sampling line? Does the RH sufficiently low such that the size measurements were not affected?

**Specified size range for the APS (page 4 line 19). Also added a discussion of the RH within the sampling line and how much that would have affected size distributions (page 4, lines 22-27).**

5. Section 2.5: I was confused about whether the CCN measurements were conducted for super-saturation scan or diameter scan. Please specify.

**Section 2.5 refers to "Size-resolved cloud condensation nuclei concentrations", states that the CCN counter was coupled with SMPS, and refers to several papers (Petters et al., 2009; Petters and Kreidenweis 2007) that were seminal in establishing size-resolved (diameter scanning) CCN measurement methods. We are unsure if we can make it more clear to the reader, but perhaps this is an issue of terminology. We added a parenthetical note that the reader may be familiar with this technique under the name "diameter scan" to page 4, line 36.**

6. Section 3.2, Page 6 Line 25: Say explicitly what correction is needed for the aethalometer. Correction for back scattering?

**Reworked sentence to make clear that the correction required is for back-scattering (page 6, line 32).**

7. Section 4.2: Report the "all-study mean" values, and the percentage differences of mean values between the PGF and CTL periods.

**Done as requested. See page 10, lines 5-8.**

8. Fig. 2: Do the colors in the wind rose plots represent the binned wind speed, as stated in the legend, or a relative probability, as stated in the caption? Also, use SI unit m s$^{-1}$ instead of kts?

**The caption of Fig. 2 has been changed to more accurately describe that the colored petals of the wind represent the relative distributions of wind speed from the given direction. The units on the wind rose petals and colorbars have been changed from kts to m s-1. Note that this changes the colorbar values slightly compared to the previous version.**

9. Fig.2 vs. Fig. 4: Fig. 4 shows that the mean value of APS particle number concentration was about one order of magnitude lower during the PGF than that during the CTL (2.4 cm-3 vs 14.9 cm-3, as mentioned in Page 10, Line 27). However, their median values were very similar, as shown in Fig. 2a. Please provide some explanation.

**Figure 2 displays the max/min/median and upper/lower quartiles of hourly APS integrated number concentration. This is a different quantity than that displayed in Fig. 4, which is the PGF/CTL**

**composite APS size distribution displayed as dlogN/dlogDp or dlogN/dlogDp. It is likely that the two quantities are drawn from different probability densities. We acknowledge that the label in Fig. 2 and in section 4.2 where the integrated quantity is referred to as "APS" is confusing. Therefore, we have changed page X, lines Y-Y to introduce the symbol $n^{APS}$ and define it. We have also changed the corresponding label in Figure 2.**

10. Section 4.4: I found the normalization of ratios not intuitive. Add an equation in the main text, or in an appendix.

**Unmodified peak ratios are dependent upon which peak is in the denominator. We utilized a normalization scheme employed by Cahill et al., (2012). The scheme is discussed further in the text at page 12, line 2.**

11. Section 4.4: Any explanation for the association between high OC:soot ratio and high abundance of amines-type particles (Fig. 8a)? Do the OC ions usually nitrogen containing for the CLT cases?

**Amines contain chemical chains of OC, so it is not surprising that the measured particles had a high OC/EC ratio. A further explanation has been added to the text at page 12, line 25.**

12. Fig. 10: I found this figure is confusing and less informative. As the values have been mentioned in the text, this figure can be eliminated. The corresponding method section (section 3.5) could be significantly shortened.

**This figure has been redacted and the corresponding methods section revised (page 9, line 10).**

*Technical comments:*
1. Page 5, Line 2: Please check the symbol (kappa?).
2. Page 7, Line 20: missing "." after "radiative effects"
3. Page 7 Line 24: Extra "." after "particle aging"

**Fixed formatting issues in manuscript.**

---

## Author Comment (AC2) · 5 Sep 2016

We would like to thank the reviewer for their valuable comments. We have reworked the paper to address the relevant issues where necessary. The reviewer comments are written in plain text, our response and changes to the manuscript in bold.

*General comments:*
As a correlation of air masses to their origin seems to be of particular importance in this work, I wonder why trajectories were not used at all. Also, I wonder if the authors could comment on the fact if their observation are particularly important to only the location of the study, or to sea side sites in general, or even to more locations worldwide. This could be discussed in a few sentences in the summary. It would also be nice if they related their results to the points they raise in the introduction (bulleted point list) in the summary, to see how their results match with the here cited literature.

**It is unclear whether Lagrangian models (e.g. HYSPLIT, FLEXPART) are able to resolve flows at the scale of Petaluma Gap Flow (horizontally ~25 km, vertically ~ 300m, temporally ~ 10 hrs), particularly if these Lagrangian models are driven by publicly available atmospheric analyses. Furthermore, previous literature has already formulated methods for determining the presence of Petaluma Gap Flow, and we do not comment on specific trace constituent sources beyond noting that they likely originate east of the Petaluma Gap. For these reasons, the use of back trajectories was not deemed necessary.**

**We've added a discussion of the relevance of this study and importance to general seaside sites to the summary (page 15, lines 29-33). We would also like to note that each hypothesis raised in the introduction is addressed individually in the Summary section starting on page 14, line 19 and concluding on page 15, line 11.**

*Specific comments:*
page 3, line 12: Maybe include a map of the area showing the mountain range (at least some of it), the gap and the sampling location.
**Added new figure 1, which is a map of the study site, relevant geographical features in CA and the Petaluma Gap Flow.**

[Figure]

page 3, line 24: Say explicitly that some more information on the instrumentation follows below. And mention explicitly when the measurements took place (which month, and for how long).
**Explicit explanations added as requested (page 3, lines 13 and 10).**

page 4, line 19: "PSL" was already introduced above (line 10 on this page), but as PSLs - homogenize and use acronyms once you have defined them.
**Changed all instances of PSL to PSLs.**

page 4, line 19: Add values for the RH that were generally observed, and the maximum values.
**Added max, min and average values of RH (page 4 starting at line 21). Also added more thorough discussion of sampling line RH over the study.**

page 4, line 25: This section, short as it is, is not only "Aerosol Mass". More correct would be "Aerosol and BC mass concentration".
**Changed section title to "Aerosol and BC Mass Concentration."**

page 5, line 2: Check the symbol in here - I guess it should have been a "kappa", but in my version, it was a very strange symbol instead.
**This appears to have been an issue with the word processing and operating system not communicating correctly. Changed to correct symbol (page 5, line 8).**

page 5, line 27: A sentence or two describing the methodology of N06 would be good, as this is the basis for a crucial part of your work.
**Added a couple sentences detailing Neiman et al methodology (page 5, line 35).**

page 5, line 34-35: I was puzzled about the use of mPGF and PGF. Maybe explain the difference between the two explicitly in a sentence where mPFG first appears. And make sure you use both consistently in the text.
**We added text explicitly discriminating mPGF from PGF on page 5, lines 26-28.**

page 7, line 2: Refer to section 3.1 (otherwise it is a self-reference).
**Changed to section 3.1 as recommended.**

page 8, line 28: What do you mean by a "dry free atmosphere"?
**Reworded to add clarity on page 9, line 36.**

page 9, line 25: Some formatting error - too large spaces between words at the end of the line.
**Fixed spacing issue.**

page 9, line 28: What do you mean by "the difference in likely concentration"? Reformulate.
**Changed "likely" to "normalized" (page 9, line 36).**

page 10, line 31: The size distributions shown in Fig. 4: Are they averages for all times of CTL and PGF, or just single distributions you used as examples? Explain in the text or caption.
**Changed in text to reflect that these are the average size distributions for all CTL periods and all PGF periods (page 10, line 31).**

page 13, top: In the text (related to Fig. 9b), it needs to be explained how CDNC were obtained. (It is not enough to mention an adiabatic parcel model in the summary.)
**We have added clarification that we are referring to the Cohard et al. method and the integrated CCN spectra as detailed in section 3.5 on page 8, lines 8-10.**

page 13, line 6 ff: Mention explicitly that the "Twomey effect albedo change" is what is shown in Fig. 10 as "fractional albedo change". I understand that in the text you try to relate the albedo measured by MODIS to the derived fractional albedo change, but I find this part of the text rather confusing. Consider rewriting this part.
**Another reviewer suggested that we redact Figure 10 and simply discuss the results in more detail in the text. We have removed the figure and rewritten the section in question with this and the other request in mind (page 13, lines 14-20).**

page 13, line 12: "entrain" is not the best choice of word (a cloud is not a thing that is moved from an air mass with CTL properties into an air mass with PGF properties), it is rather that the clouds form in the air mass, which has a certain aerosol in it. Reformulate.
**Changed from entrain to "condense on". See page 13, line 16.**

page 14, line 13: When you first mention aged SSA, "aged" is not capitalized, but it is capitalized here and in other locations. Be coherent.

**Capitalized all instances of "aged" (in reference to the Aged SS type).**

page 14, line 27: Remove the extra ".".
**Removed the extra ".".**

Fig. 2 and 3: The change in color code is confusing (blue is "all" in Fig. 2, but CTL in Fig. 3, red is PGF in both, and black is CTL in Fig. 2 but local in Fig. 3). Please change this, and keep the colors for CTL and PGF for all figures where possible and applicable (e.g., Fig 4 and Fig. 9).
**Changed relevant figures (3, 4, 5, and 10) to incorporate a color scheme such that:**
**LOCAL = BLACK**
**PGF = RED**
**CTL = BLUE**
**ALL = GREEN**

Fig. 5: Explain that the explanation of the abbreviations can be found in Tab. 2.
**Added text to caption to explain that the abbreviations can be found in table 2.**